# Artificial Neural Networks Generate Human-like Continuous Speech Perception

**Gasser Elbanna**
Speech and Hearing Bioscience and Technology
Harvard University
Boston, MA 02115
gelbanna@mit.edu

**Josh H. McDermott**
Department of Brain and Cognitive Sciences
Massachusetts Institute of Technology
Cambridge, MA 02139
jhm@mit.edu

## Abstract

Humans have a remarkable ability to convert acoustic signals into linguistic representations. To advance toward the goal of building biologically plausible models that replicate this process, we developed an artificial neural network trained to generate sequences of American English phonemes from audio processed by a simulated cochlea. We trained the model with phoneme transcriptions inferred from text annotations of speech corpora. To compare the model to humans, we ran a behavioral experiment in which humans transcribed non-words, and evaluated the model on the same stimuli. While humans slightly outperformed the model, the model exhibited human-like patterns of phoneme confusions for consonants (r=0.91) and vowels (r=0.87). Additionally, the recognizability of individual phonemes was highly correlated (r=0.93) between humans and the model. These results suggest that human-like speech perception emerges from optimizing for phoneme recognition from cochlear representations.

## 1 Introduction

The core computational challenge of speech perception is the absence of consistent one-to-one mappings between the acoustic signal and the sub-lexical units (such as phonemes) that make up speech (1; 2). Despite substantial acoustic variability across speakers, speaking rates, and environmental conditions, humans exhibit remarkable accuracy in perceiving speech sounds. Various theoretical ideas have been proposed to account for our abilities. These include acoustic invariances (3), categorical perception (4), and the motor theory of speech perception (5), among others. While these theories have guided research, they are not sufficiently specified to fully explain how listeners achieve such robust speech recognition.

In parallel, computational models of speech perception have been developed to explain aspects of speech perception. Traditional models often operated on abstract representations or handcrafted features, and were generally unable to account for human recognition of real-world speech (6; 7; 8; 9). In recent years, artificial neural networks (ANNs) have emerged as a powerful alternative, learning directly from raw acoustic input. These models are often able to approach human levels of performance, unlike traditional models, and have helped explain human-like perceptual abilities in other domains of audition (10; 11). However, most available speech ANN models lack biological plausibility and often do not exhibit the same patterns of performance as human listeners (12; 13; 14).

Some work has attempted to bridge this gap by developing biologically plausible speech models, for instance using a simulated cochlear representation as a front-end for a neural network (15; 16). These models exhibit human-like patterns of speech intelligibility in some conditions but do not perform continuous speech recognition because they were trained to recognize a set of words substantially smaller than the full vocabulary of English, and thus cannot account for many aspects of human speech perception.

38th Conference on Neural Information Processing Systems (NeurIPS 2024).

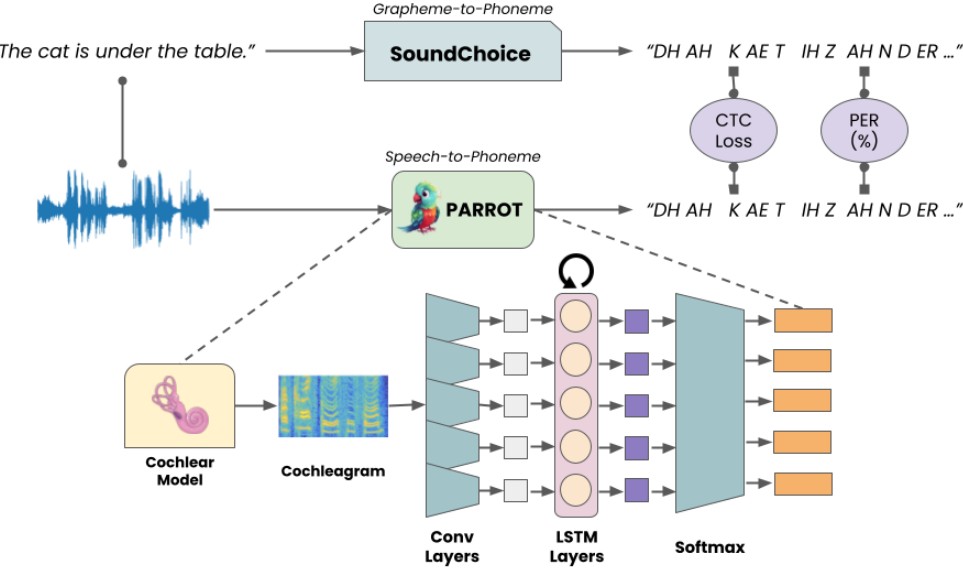

Figure 1: PARROT architecture and training pipeline.

In this work, we present a novel speech model called PARROT that is trained to generate sequences of English phonemes from simulated cochlear input. We directly compare the model's performance to that of human listeners using a non-word transcription task. This task allows us to analyze and compare patterns of successes and failures of phoneme recognition, shedding light on the similarities and differences between human and model performance and providing a first step towards more realistic models of continuous speech perception.

## 2  Methods

### 2.1  Model Architecture and Task Objective

The first stage of the architecture is a cochlear model adapted from prior work (17) that processes speech waveforms sampled at 16 kHz. The simulated cochlea applies gamma-tone filters with center frequencies between 40 Hz and 20 kHz, with tuning and spacing intended to replicate that of the human ear. The output from the filter banks is half-wave rectified and low-pass filtered with a cutoff frequency of 4 kHz, simulating the upper limit of phase-locking (18). The filtered output is then downsampled to a sampling rate of 8 kHz and raised to the 0.3 power to replicate the nonlinear amplification of the ear (19).

Cochlear representations are fed into six 2-dimensional convolutional layers with 512 channels, which further downsample the signal from 8 kHz to 50 Hz, encoding 20 ms of speech per frame. After each convolutional layer, batch normalization is applied, followed by a ReLU non-linear activation function. The resulting latent representations are passed through six bi-directional Long Short-Term Memory (LSTM) layers with hidden size of 512, which capture temporal dependencies across frames. Finally, the LSTM hidden states are projected into a 40-class phoneme space (consisting of 39 phonemes and a blank class) using a linear fully connected layer. The logits are converted into a probability distribution via a softmax function.

The model was trained to map the probability distribution of phoneme classes to tokens using a Connectionist Temporal Classification (CTC) loss (20) (see Section A.1). Phoneme Error Rate (PER) was calculated after aligning predicted and ground truth phonemes using the Levenstein distance algorithm (21). The percentage of phoneme errors were computed as in Eq. 1.

$$PER(\%) = \frac{No.\,of\,Insertions + No.\,of\,Deletions + No.\,of\,Substitutions}{Total\,No.\,of\,Ground\,Truth\,Phonemes} \tag{1}$$

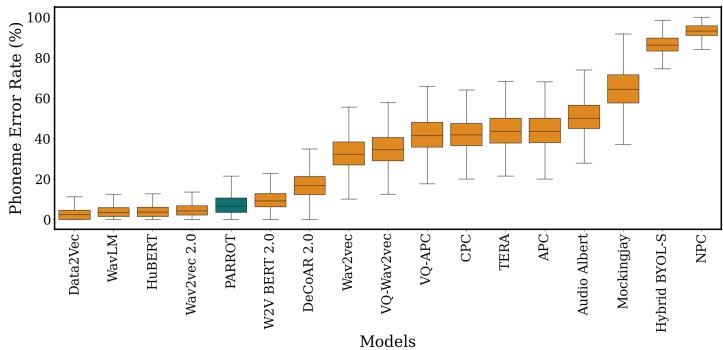

Figure 2: Phoneme Error Rate (%) of speech models on Phoneme Recognition task from SUPERB Benchmark. PARROT result is shown in teal. The lower the value, the better the model.

## 2.2 Experimental Setup and Evaluation

Because there is little available phoneme-labeled speech data, we used a pseudo-supervised training approach, employing a Grapheme-to-Phoneme model called SoundChoice (22) to transcribe phonemes from text annotations of large-scale speech corpora. We transcribed around 6 Million utterances from open-source corpora including GigaSpeech (23), Librispeech (24), VCTK (25), LJSpeech (26), Speech commands (27), FSDD (28), and TIMIT (29), yielding around 10,000 hours of training data. We used the Librispeech dev set for validation and the Librispeech test set for testing which were not part of the training data. Also, we benchmarked the model on SUPERB phoneme recognition task (30).

## 2.3 Non-word Recognition Experiment

To compare model and human speech perception, we designed a non-word recognition task where participants transcribed synthesized non-words. The non-words were generated with Wuggy (31), a pseudo-word generator that produces non-word variants from real words while abiding by English phonotactics rules. We generated around 15,000 non-words and selected a subset of 5,000 that maximized representation of the least common phonemes. Non-words were then synthesized using the MeloTTS text-to-speech model (32).

We ran an online behavioral experiment on Prolific. Each of the 100 participant transcribed 200 non-words randomly chosen from the pool of 5000 non-words. Participants had to first pass a headphone check (33). We converted each text string response into phonemes using the same G2P model used to generate training data labels. The same non-words were presented to the model.

## 3 Results

### 3.1 Model Evaluation

The model achieved a median PER of 6.8% on on Librispeech test set, and performed competitively on a standard phoneme recognition benchmark (30) (Figure 2).

### 3.2 Human-Model Comparison

We computed the PER for each non-word in the experiment. The model was slightly worse than humans (median PER of 33% vs 29%; Figure 3.a). However, individual phonemes varied in the accuracy with which they were recognized, and the phoneme-wise accuracy was highly correlated between humans and the model (r=0.92; p<0.01; Figure 3.b-c). The correlation remained high when calculated separately for consonants (r=0.97, p<0.01) and vowels (r=0.86, p<0.01).

To assess whether the model replicated the pattern of confusions exhibited by humans, we compared human and model confusion matrices, separately for consonants and vowels (Figures 3.d-i). The off-diagonal matrix entries were also strongly correlated between humans and PARROT for both consonants (r=0.91, p value<0.01) and vowels (r=0.87, p value<0.01). The same phoneme was

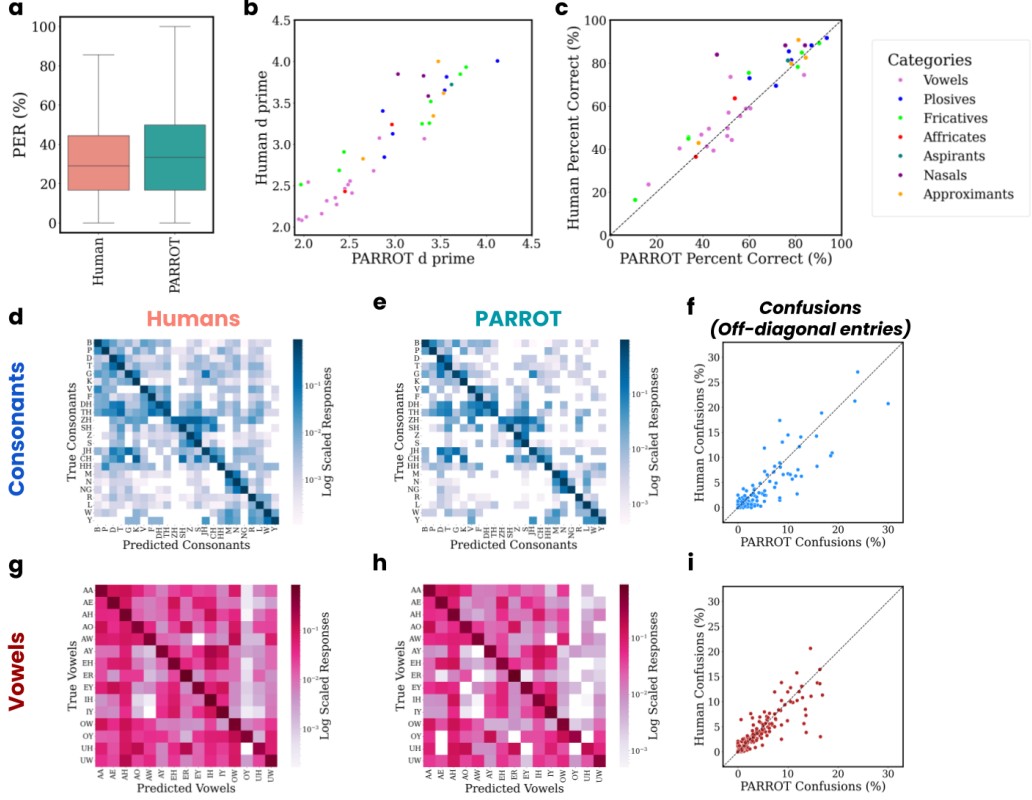

Figure 3: Human-Model comparison on non-word recognition task. (a) PER distribution for Humans and PARROT across non-word stimuli. (b) Human vs. model recognition accuracy for individual phonemes, expressed as d prime. Colors indicate manner of articulation. (c) Same as b, but plotting percent correct. (d) and (g) Phoneme confusions in humans for consonants and vowels, respectively. (e) and (h) Phoneme confusions in PARROT for consonants and vowels, respectively. (f) and (i) Off-diagonal correlation between humans and PARROT for consonants and vowels, respectively.

confused the most for both humans and PARROT (/ZH/, as in "measure"). Similarly, the same phoneme was confused the least for both (/K/ as in "cat").

## 4 Discussion

Compared to existing automatic speech recognition systems, the model demonstrated competitive performance on unseen data and various transcription methods (see Figure 2). Humans performed slightly better than the model on the non-word recognition task. However, at the phoneme level, the model exhibited a similar pattern of phoneme confusions as humans, both for consonants (r=0.91) and for vowels (r=0.87). The recognizability of individual phonemes was also highly correlated between humans and the model (r=0.93), highlighting the model's alignment with human perception.

## 5 Conclusion

We developed a novel deep learning model that was trained to recognize phonemes using data transcribed by an existing Grapheme-to-Phoneme model. The model performed competitively on the task and showed human-like patterns of phoneme confusions. The findings collectively suggest that aspects of human-like speech perception emerges by optimizing for phoneme recognition from cochlear representations. In future work, we plan to identify the key model components driving this alignment. The results provide a first step towards building biologically-plausible models that replicate and explain human speech representations.

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

## A Appendix / supplemental material

### A.1 Training Details

Training examples varied in duration, so we padded each example in a batch to the length of the longest duration in the batch. We used batches of size 4 with 2 gradient accumulation steps. We trained the model on 8 A100 GPUs yielding an effective batch size of 4*2*8 = 64. The model was trained on a total of 400,000 gradient steps per GPU translating into a total of 5 epochs. We use Adam optimizer (34) with weight decay of 0.01 and a warming up the learning rate (LR) linearly for the first 10,000 steps reaching a peak of 0.001. LR is fixed to peak value until training reaches 200,000 steps then LR decreases using cosine annealing for the second half of training reaching min LR of 0.00001.

