# OpenReview forum: "Artificial Neural Networks Generate Human-like Continuous Speech Perception"
_NeurIPS.cc/2024/Workshop/UniReps — UniReps_

### Official Review · Reviewer_5V8h · 2024-10-01
**Nice work on cochear models and AI**

**Rating:** 7
**Confidence:** 3

**Review:**

This paper introduces PARROT, an artificial neural network model designed to mimic human speech perception by generating sequences of phonemes from simulated cochlear input. T

he architecture combines a biologically inspired front-end with deep neural network components, showing promising alignment with human patterns of phoneme recognition.

I liked the experimental evaluation, including comparisons to human performance on non-word transcription tasks. This is thorough and indicates strong model-human correlation.

However, the paper could benefit from a more in-depth discussion of limitations and future work, particularly regarding generalizability and improvements in biological plausibility. Additionally, the societal impacts and potential applications are underexplored, but this could be due to page limit constraints so I think it's fine.

 Overall, the paper makes a solid contribution to biologically plausible speech perception models, and with minor improvements in framing and discussion, it would be well-suited for presentation at UNIREPS workshop NeurIPS.

---

> ### Author Response · Authors · 2024-10-23
> **Response to Reviewer 5V8h**
>
> The authors thank reviewer 5V8h for their comments. We acknowledge the importance of providing more in-depth discussion and future work. We will try to incorporate this feedback within the given page limits for the camera-ready version.

---

### Official Review · Reviewer_SYbj · 2024-10-03
**Interesting human-model comparison but lacks proper experimental validation and makes overstated claims.**

**Rating:** 3
**Confidence:** 4

**Review:**

The paper presents PARROT, a speech recognition model that aims to bridge the gap between biologically plausible models and automatic speech recognition (ASR) systems by incorporating a cochlear front end and comparing its performance with human listeners.

While the work presents interesting human-model comparisons on non-word recognition, there are several significant concerns, mainly regarding experimental design issues:

- The model is trained on 10,000 hours of data (Section 2.2), but comparisons with other models on the Phoneme Recognition task from the SUPERB Benchmark lack clarity about training data differences. To the best of my knowledge, all other models only used 100 hours of labeled data from the LibriSpeech train-clean-100 subset, as validated in the SUPERB paper [1].

- There is an absence of ablation studies to validate the importance of the cochlear front end. Also, the paper shows limited comparative analysis with existing biologically-inspired speech recognition approaches. The lack of comparative analysis makes it impossible to assess the true advancement of their approach over existing techniques in the field. The absence of proper and comparable baselines, ablation studies, or comparative experiments significantly weakens the paper's empirical contribution.

- Line 101: "unseen data (Figure 2)". Since Librispeech is used in training (line 68), testing on the Librispeech test set is not to be considered "unseen."

- The paper's title claims to 'explain' continuous speech perception in humans, which appears overstated given the evidence presented. Additionally, the abstract could more clearly articulate the specific contributions and advances over prior work.

In conclusion, while the human-model comparison presents interesting results, the paper's experimental validation is insufficient to support its broader claims about explaining human speech perception. Since the model also uses more labeled data in its training, it is not convincing that it can perform competitively with existing techniques.

Reference:
[1] Shu-wen Yang, Po-Han Chi, Yung-Sung Chuang, Cheng-I Jeff Lai, Kushal Lakhotia, Yist Y. Lin, Andy T. Liu, Jiatong Shi, Xuankai Chang, Guan-Ting Lin, Tzu-Hsien Huang, Wei-Cheng Tseng, Ko-tik Lee, Da-Rong Liu, Zili Huang, Shuyan Dong, Shang-Wen Li, Shinji Watanabe, Abdelrahman Mohamed, Hung-yi Lee. SUPERB: Speech processing Universal PERformance Benchmark. INTERSPEECH 2021.

---

> ### Author Response · Authors · 2024-10-23
> **Response to Reviewer SYbj**
>
> The authors thank reviewer SYbj for their comments. Below we address the comments raised by the reviewer.
>
> * This is a misunderstanding. The reviewer conflated pre-training data and data used for the SUPERB downstream task. Regarding pre-training data, all models were pre-trained on at least **1,000** hours of data. Some of them were trained on **60,000** hours of speech data. Whereas, all models were evaluated on phoneme recognition (i.e., SUPERB benchmark) using the same amount of data.
>
> * We acknowledge the importance of baseline comparisons. This work is on-going and a baseline will be added in future work.
>
> * The test set of Librispeech **was not** part of the PARROT training. This test set has different utterances and talkers that are not part of the train set or dev set in Librispeech. Thus, it is in fact unseen data. We will clarify this in the camera-ready version.
>
> * We will change the title to *“Artificial Neural Networks Generate Human-like Continuous Speech Perception”*.

---

### Official Review · Reviewer_LxEn · 2024-10-05
**Questioning the Trade-Off Between Interpretability and Performance in the PARROT Model**

**Rating:** 6
**Confidence:** 5

**Review:**

The paper mentions that "Humans have a remarkable ability to convert acoustic signals into linguistic representations," and that "the recognizability of individual phonemes was highly correlated (r=0.93) between humans and the model." However, from Figure 2, we can observe that the PARROT method, represented by the teal-colored box plot, ranks among the lower-performing models in terms of phoneme recognition. This raises an important question: does this imply that the human ability to convert acoustic signals into linguistic representations is not as "remarkable" when compared to other models?

Even though PARROT's phoneme recognition is said to closely mimic human capabilities, it still falls behind the more advanced models in terms of overall performance. So, if the PARROT model and human abilities both perform below the leading models, why should we prioritize developing an interpretable model that mimics human-like processes? The lower performance seems to limit the practical applications of this approach, even though it offers higher interpretability.

In light of this, I believe the paper should address the trade-off between interpretability and performance more explicitly. While biologically plausible models like PARROT may offer valuable insights into human cognition, their practical utility might be constrained if their performance does not match up to more advanced, less interpretable models.

---

> ### Author Response · Authors · 2024-10-23
> **Response to Reviewer LxEn**
>
> This is a misunderstanding. Figure 2 shows the **Phoneme Error Rate (PER)** for all models. Thus, the **lower** the value, the **better** the model. Consequently, PARROT is one of the **best-performing** models on the task of phoneme recognition. There is no trade-off to be addressed pertaining interpretability and performance. In the camera-ready version, we will explicitly write on the y-axis Phoneme Error Rate instead of PER to make this issue more obvious to the reader.

---

### Decision · Program_Chairs · 2024-10-10

**Decision:**

Accept

**Comment:**

In light of the reviewers' feedback and relevancy of the submission, we are pleased to accept this paper for presentation at UniReps 2024. We kindly ask the authors to incorporate the reviewers' suggestions and feedback in the final camera-ready version of the manuscript.